# COVID-19: Unveiling the Neuropsychiatric Maze—From Acute to Long-Term Manifestations

**DOI:** 10.3390/biomedicines12061147

**Published:** 2024-05-22

**Authors:** Daniela Ariza, Lily Castellar-Visbal, Maria Marquina, Diego Rivera-Porras, Nestor Galbán, Raquel Santeliz, Melissa Gutiérrez-Rey, Heliana Parra, José Vargas-Manotas, Wheeler Torres, Laura Quintana-Espinosa, Alexander Manzano, Lorena Cudris-Torres, Valmore Bermúdez

**Affiliations:** 1Endocrine and Metabolic Diseases Research Center, School of Medicine, Universidad del Zulia, Maracaibo 4002, Venezuela; arizathings@gmail.com (D.A.); marquinastagg@gmail.com (M.M.); nestorag17@gmail.com (N.G.); gabrielasanteliz29@gmail.com (R.S.); helianapp@hotmail.com (H.P.); wheelertorres16@gmail.com (W.T.); amanzano_8@hotmail.com (A.M.); 2Universidad Simón Bolívar, Facultad de Ciencias de la Salud, Barranquilla 080001, Colombia; lily.castellar@unisimon.edu.co (L.C.-V.); melissa.gutierrez1@unisimon.edu.co (M.G.-R.); jose.vargas@unisimon.edu.co (J.V.-M.); laura.quintana@unisimon.edu.co (L.Q.-E.); 3Universidad Simón Bolívar, Facultad de Ciencias Jurídicas y Sociales, Centro de Investigación en Estudios Fronterizos, Cúcuta 540001, Colombia; diego.rivera@unisimon.edu.co; 4Departamento de Ciencias Sociales, Universidad de la Costa, Barranquilla 080001, Colombia; lcudris3@cuc.edu.co; 5Universidad Simón Bolívar, Facultad de Ciencias de la Salud, Centro de Investigaciones en Ciencias de la Vida, Barranquilla 080001, Colombia

**Keywords:** SARS-CoV-2, COVID-19, neuroinflammation, encephalitis, stroke, Guillain–Barré syndrome

## Abstract

The SARS-CoV-2 virus has spread rapidly despite implementing strategies to reduce its transmission. The disease caused by this virus has been associated with a diverse range of symptoms, including common neurological manifestations such as dysgeusia, anosmia, and myalgias. Additionally, numerous cases of severe neurological complications associated with this disease have been reported, including encephalitis, stroke, seizures, and Guillain–Barré syndrome, among others. Given the high prevalence of neurological manifestations in this disease, the objective of this review is to analyze the mechanisms by which this virus can affect the nervous system, from its direct invasion to aberrant activation of the immune system and other mechanisms involved in the symptoms, including neuropsychiatric manifestations, to gain a better understanding of the disease and thus facilitate the search for effective therapeutic strategies.

## 1. Introduction

The COVID-19 pandemic, caused by the SARS-CoV-2 virus, has resulted in a significant global public health crisis. By March 2024, the virus had demonstrably impacted global life expectancy and claimed the lives of 7,043,660 people worldwide, according to the WHO Health Emergencies Programme [1]. The initial focus of the COVID-19 pandemic lay in its acute phase, causing severe respiratory illness and death. Nevertheless, as scientific understanding evolves, a growing concern is the transition to long-term COVID-19 following the initial acute SARS-CoV-2 infection [2,3]. A growing body of evidence suggests a parallel with other viral illnesses, with a spectrum of persistent and novel symptoms emerging [4]. These sequelae, encompassing post-COVID-19 syndrome, long-COVID-19, and post-COVID-19 conditions, are characterized by their onset during or after the documented infection and persistence beyond 12 weeks from diagnosis [5]. This condition, characterized by organ dysfunction, and neurological and psychiatric complications, affects a substantial proportion of COVID-19 survivors and represents a significant long-term public health challenge [6,7,8].

The SARS-CoV-2 infection begins with the entry of the virus through the upper respiratory tract (Figure 1 and Figure 2). Angiotensin-converting enzyme 2 (ACE2) is the functional receptor for SARS-CoV-2, found in type II alveolar cells, as well as extrapulmonary tissues such as the heart, ileum, brain, kidney, bladder, and the endothelium [9,10]. This fact may explain other manifestations related to SARS-CoV-2, including thrombosis [11], myocarditis [12], diarrhea [13], renal [14], and hepatic failure [15]. In this regard, the 2019 coronavirus disease (COVID-19) is associated with a range of diverse symptoms, ranging from mild infection, in the majority of cases, to more severe conditions such as pneumonia, acute respiratory distress, and multi-organ failure, which occur more frequently in patients with risk factors, and have a mortality rate of 3% to 4% [16,17].

The typical manifestations have been extensively described, including fever (98%), cough (76%), dyspnea (55%), nausea/vomiting or diarrhea (31%), and fatigue (44%). However, it is evident that neurological symptoms associated with COVID-19 go beyond simple olfactory (41%) and gustatory (38.2%) dysfunctions, even in patients without prior respiratory pathology [13,18,19,20,21]. In this vein, patients with severe illness have presented neurological manifestations and complications in a higher proportion [22], such as acute cerebrovascular events, altered consciousness, encephalitis, and skeletal muscle injury [23,24]. Furthermore, other neurological complications included Guillain–Barré Syndrome (21.8%), followed by stroke (16.4%) and optic neuritis (12.7%) [25]. Likewise, the complications could lead to disabling conditions such as multiple sclerosis, acute hemorrhagic necrotizing encephalitis, Alzheimer’s, and Parkinson’s, among others, likely due to the cytokine storm induced by severe COVID-19 infections [26,27].

This review aims to analyze the acute and late neuropsychiatric complications of COVID-19 and their clinical practice implications, as they have increasingly been recognized as an important aspect of the disease, ranging from loss of smell and taste to neuropsychiatric manifestations such as psychosis, providing an update on the potential underlying mechanisms that led to their presentation.

## 2. Central Nervous System (CNS) Invasion and Damage Mechanisms by COVID-19

As the number of COVID-19 cases grows, its neurological manifestations are becoming more recognized. They range from relatively benign symptoms, such as anosmia and dysgeusia, to life-threatening manifestations, such as cerebral infarcts or encephalitis [28]. It has even been suggested that there exists a pattern of neuropathological alterations in patients with COVID-19 who presented severe pneumonia characterized by hypoxia, neocortical infarctions, and small hemorrhagic and non-hemorrhagic lesions in the white matter [29,30,31,32]. In addition, a study by Kanberg et al. reported evidence of neuronal injury and glial activation in patients with COVID-19 [33]. Despite this, the mechanisms through which SARS-CoV-2 affects the nervous system remain unclear. Multiple possibilities are being considered, including virus invasion of nervous system cells through neurotropic dissemination, hematogenous spread, and aberrant immune system activation [34].

### 2.1. Direct Invasion by the SARS-CoV-2 Virus

Many viruses have neuroinvasive potential in humans, including the Coronaviridae family. It has been found that several members of this family can invade the nervous system, both in animal models and in humans (Figure 3) [35,36,37,38]. In an in vitro study using a 3D human cerebral microphysiological system derived from induced pluripotent stem cells, consisting of differentiated mature neurons and glial cells (astrocytes and oligodendrocytes) [39], conducted by Bullen et al., invasion and replication of SARS-CoV-2 were observed in different cells in the nervous system [40]. Additionally, subsequent studies have found that these neuronal infections lead to cell death with a loss of excitatory synapses [41]. Another relevant finding obtained through this type of organoid is the variability in the replication of different strains of SARS-CoV-2 in the central nervous system (CNS). According to the findings of Hou et al., Omicron BA.2 was the variant that replicated most efficiently, compared to the wild type (WT), Delta, and Omicron BA.1. According to this study, this difference may be due to a lower induction of type I interferons (IFN-α and IFN-β) and pro-inflammatory cytokines (TNF-α and IL-6) by BA.2 [42]. However, these models do not encompass all the conditions in the brain and in vivo results may differ.

However, other studies have observed genetic material from SARS-CoV-2 in the brain tissue and cerebrospinal fluid in patients with COVID-19 [43,44], which supports the ability of this new virus to invade the CNS. Additionally, through single-cell RNA sequencing (scRNA-seq), it was evidenced that precursor cells of oligodendrocytes and astrocytes in the substantia nigra and cerebral cortex exhibit a high expression of ACE2, and to a lesser extent, TMPRSS2 [45]; these proteins are crucial for the virus to enter the cell, acting as receptor and priming agent, respectively, for the spike protein (S) of SARS-CoV-2 [46]. Therefore, the substantia nigra and cerebral cortex are defined as high-risk tissues for infection by this virus [45].

Neurotropic dissemination is possible regarding the pathway through which this pathogen enters the nervous system. This step involves the infection of peripheral neurons whose intracellular transport machinery is exploited by the virus to gain access to the CNS [47,48]. It has been speculated that SARS-CoV-2, like other viruses, may invade the brain through olfactory pathways [49], as suggested by a case report of a 24-year-old woman with COVID-19 who presented anosmia among her symptoms and whose brain MRI showed cortical hyperintensity in the right rectal gyrus and subtle hyperintensity in the olfactory bulbs, findings consistent with viral brain invasion [50].

This hypothesis is supported by a study conducted by Meinhardt et al., in which autopsies of 33 patients with COVID-19 were performed. Using real-time quantitative reverse transcription polymerase chain reaction (RT-qPCR), abundant SARS-CoV-2 copies were found in the olfactory mucosa, directly beneath the ethmoid’s cribriform plate, in 20 out of 30 individuals; viral RNA copies were also found, in lower quantities, in the olfactory bulb in 3 out of 31 people. Similarly, the cornea, conjunctiva, and oral mucosa showed viral RNA in smaller amounts, positioning the oral and ophthalmic routes as other potential entry points to the CNS [51].

Notably, in a study conducted by Brann et al., ACE2 expression was observed in supporting cells, stem cells, and perivascular cells of the olfactory epithelium and olfactory bulb through scRNA-seq; ACE2 expression was not found in neurons of these tissues [52]. On the contrary, in another study conducted by Nampoothiri et al., olfactory and vomeronasal sensory neurons, at least in human embryos, expressed high levels of both ACE2 and TMPRSS2 [53]. Therefore, the presence of this protein in the neurons of the olfactory epithelium is still a subject of debate. However, its absence does not seem to preclude viral invasion completely, as there is evidence of viral infection and replication in cells that do not express this molecule, such as pericytes, in which SARS-CoV-2 would use Natriuretic Peptide Receptor 1 (NRP1) [54]. Similarly, in the olfactory bulb of adult individuals, a high expression of sialic acid and integrins are potential receptors for the S protein of SARS-CoV-2 [53,55,56]. However, studies analyzing the virus’s possible spread through the olfactory pathway are still limited, making it necessary to investigate this aspect for a more precise understanding.

Another potential neuroinvasive mechanism is hematogenous transmission, which involves the entry of viral particles from the blood into the CNS. While the blood–brain barrier (BBB) is an immunological mechanism that hinders CNS invasion, various viruses have found ways to breach it, either by infecting endothelial cells of the BBB [57] or through the infection of leukocytes that subsequently migrate to the CNS [58]. In this regard, Paniz-Mondolfi et al. found, during the post-mortem examination of a 74-year-old man with Alzheimer’s disease and infected with SARS-CoV-2, viral particles in neuronal bodies and endothelial cells from frontal lobe sections using electron microscopy, where blister-like structures of virus-like particles were observed entering or exiting the endothelial wall. This finding indicates transcellular penetration through the endothelial cells in the cerebral microvasculature [59]. In this regard, a study by Proust et al., in which they infected cells (primary human pericytes, fetal astrocytes, endothelial cells, and a microglial cell line) and a 3D model of the blood–brain barrier with variants of SARS-CoV-2, WT, Alpha, Beta, Delta, Eta, or Omicron (BA.1), found evidence of neurological infection in all strains. However, greater neuropathogenicity was observed in WT and Omicron, demonstrating alterations in the integrity of the endothelial barrier caused by these variants, specifically in WT infection, an increase in BBB permeability [54].

Another study supporting BBB disruption in neurological manifestations caused by COVID-19 is the work of Bonetto et al. In this study, a sample of patients with Neuro-COVID (n = 79), ICU COVID (n = 78), healthy controls (n = 20), and patients with Amyotrophic Lateral Sclerosis (ALS) was taken. The findings revealed that in patients with severe COVID, markers of BBB disruption are elevated at levels comparable to or higher than those found in individuals with ALS. Similarly, in Neuro-COVID, there is an increase in matrix metallopeptidase 9 (MMP-9) and neurofilament light chain (NFL), which are associated with BBB alteration and neuronal damage, compared to patients with ICU COVID [60].

### 2.2. Aberrant Activation of the Immune System

A significant factor implicated in the pathophysiology of COVID-19 is the aggressive inflammatory response due to the overproduction of inflammatory cytokines (Figure 4).

Patients infected with SARS-CoV-2 exhibit a significant elevation in interleukin-1β (IL-1β), interleukin-1 receptor antagonist (IL-1RA), interleukin-7 (IL-7), interleukin-8 (IL-8), interleukin-9 (IL-9), interleukin-10 (IL-10), fibroblast growth factor (FGF), granulocyte colony-stimulating factor (GCSF), granulocyte–macrophage colony-stimulating factor (GMCSF), interferon-gamma (IFNγ), interferon gamma-induced protein 10 (IP-10), monocyte chemoattractant protein 1 (MC-P1), macrophage inflammatory protein-1α (MIP-1α), macrophage inflammatory protein-1β (MIP-1β), platelet-derived growth factor (PDGF), tumor necrosis factor α (TNFα), and vascular endothelial growth factor (VEGF) [18]. This cytokine profile reflects the pro-inflammatory response aimed at eliminating viral particles from the body and the anti-inflammatory response attempting to control the damage caused by this immune hyperactivity [62].

Furthermore, the cytokine storm has been associated with the severity of the disease according to observations from the study conducted by Huang et al. with a sample of 41 patients with COVID-19. Patients requiring ICU admission had higher concentrations of GCSF, IP-10, MC-P1, MIP-1α, and TNFα compared to those who did not require ICU admission [18]. Similarly, Ruan et al. found that individuals who died from COVID-19 complications had higher serum levels of C-reactive protein (CRP), interleukin-6 (IL-6), and ferritin, suggesting an underlying hyperinflammatory process [63].

COVID-19 has also been linked to nucleotide-binding oligomerization domain-like receptor containing pyrin domain 3 (NLRP3) inflammasome activation, which may be involved in both the cytokine storm and neurological manifestations of COVID-19. This complex is a common link between this disease and autoimmune diseases, like multiple sclerosis. However, the role of NLRP3 in neurological manifestations is still not fully understood [64]. Additionally, there has been an increase in the cytokine storm inducer peptidyl-prolyl cis-trans isomerase A (PPIA) in both neuro-COVID and classic severe COVID (in ICU), but it is worth mentioning that in neuro-COVID, the levels were lower. Furthermore, higher levels of PPIA and IL-10 were observed in the acute phase of ICU patients with COVID-19. Despite this difference, patients with neuro-COVID had higher levels of MMP-9 and NFL, markers of neurological damage [60].

The systemic inflammation induced by the cytokine storm mentioned can directly harm the central nervous system (CNS) via cytokines, chemokines, and soluble inflammation mediators accessing CNS through the blood–brain barrier (BBB), disrupting homeostasis and leading to cerebral inflammation, which may have a causal relationship with the neurological manifestations of the disease [65]. Furthermore, not only can an exaggerated immune response be responsible for the neurological manifestations of COVID-19, but an autoimmune response can also develop. This pattern was reported in a case report involving eleven Guillain–Barré Syndrome (GBS) patients. The improvement of some patients after the administration of Intravenous Immunoglobulin (IVIg) and the presence of anti-ganglioside GD1b antibodies in one of the cases suggest that immune response triggered by SARS-CoV-2 is involved, similar to other virus-induced GBS [66,67].

Similarly, cases of auto-antibody-mediated encephalitis have been reported in patients with COVID-19 [68,69,70] or following vaccination [71,72], as well as myelopathy with antibodies against oligodendrocytic glycoprotein (MOG) after SARS-CoV-2 infection [73]. Implicated antibodies include anti-N-methyl-D-aspartate receptor (NMDAR) antibodies, MOG antibodies, anti-amphiphysin antibodies, and antibodies against contactin-associated protein-like 2 (Caspr2) [74]. However, the available evidence regarding the production of autoantibodies implicated in neurological manifestations of COVID-19 is limited, and the relationship between these antibodies and the condition remains unclear, requiring further research in the future [75].

Another theory related to the aberrant activation of the immune system is the increase in bradykinin, an active peptide in the kallikrein–kinin system, with potent vasodilatory activity contributing to hypotension in the systemic circulation. This increase may occur due to the decreased availability of angiotensin-converting enzyme 2 (ACE2), which is depleted by SARS-CoV-2 within the epithelial cells of the lungs, leading to an inability to degrade the bradykinin analogue, des-Arg9-BK, which is associated with lung injury and inflammation [76]. All of these phenomena contribute to dysfunction in the kinin–kallikrein, renin–angiotensin, and coagulation systems, which may be implicated in a wide range of complications and manifestations of COVID-19 [77,78], such as acute respiratory distress syndrome, cytokine storm, and multi-organ failure [79].

## 3. Neurological Manifestations and Complications

Understanding the potential mechanisms by which SARS-CoV-2 may affect the nervous system, it is not surprising that in patients with COVID-19, in addition to relatively benign neurological manifestations, there are also serious complications that affect both the central and peripheral nervous system, as demonstrated by various case reports and epidemiological studies (Table 1).

### 3.1. Central Nervous System

#### 3.1.1. Encephalitis

Encephalitis involves inflammation of the brain parenchyma caused by various factors that trigger an immune response, with viruses being the most common aetiology. While coronaviruses primarily affect the respiratory system in humans, they also can affect the CNS and lead to encephalitis, as demonstrated in some cases of MERS-CoV [87] and a study conducted by Yuanyuan et al., where out of 183 children hospitalized for CNS infection, 12% had coronavirus infection [88].

Regarding COVID-19, there have been reported cases of encephalitis and meningoencephalitis associated with this disease [44,83]. In a meta-analysis by Siow et al. that included 23 studies reporting findings from 129,008 patients with COVID-19, the average incidence of encephalitis was 0.215%, which rose to 6.7% in severely ill patients [89]. Among the reported cases of patients with COVID-19 who developed encephalitis is a 60-year-old man who was admitted due to severe alteration in consciousness. He tested positive for SARS-CoV-2 through PCR from a nasopharyngeal swab. Inflammatory findings were observed in the cerebrospinal fluid (CSF) upon admission, such as mild lymphocytic pleocytosis and moderate protein elevation in the CSF. However, the CSF PCR did not detect the presence of SARS-CoV-2. Clinical improvement was achieved through high-dose intravenous steroid treatment (methylprednisolone 1 g/day for five days) [83].

While in the case mentioned above, the virus was not found in the CSF, this does not rule out the possibility of direct viral infection of the CNS, as the viral load in this fluid may be undetectable or only present transiently [83]. This finding is supported by another case report, where a 24-year-old male patient with COVID-19 and signs of meningoencephalitis showed the presence of the virus in the CSF through PCR [44].

Recently, Zhang et al. performed an autopsies study on nine critically ill patients with COVID-19 and brain samples from control subjects consisting of nine healthy deceased individuals of the same age without neurological diseases. Monocytic encephalitis associated with COVID-19 was observed, characterized by monocytes and microglial activation. Elevated levels of IL-4, IL-6, IL-8, IL-12, TGF-β, and TNF-ɑ were found in the brains of patients with COVID-19. However, neither SARS-CoV-2 mRNA nor viral proteins were detectable in the neurons and glial cells [90].

There is also increasing evidence of post-COVID-19 autoimmune antibody-mediated encephalitis, with elevated levels of these antibodies [68,69,70], which may also apparently be induced by anti-COVID vaccines as an infrequent complication. Therefore, there is no doubt that the benefits of COVID-19 vaccination outweigh the risks of neurological complications [72,91]. Another possibility is that brain inflammation results from the cytokine storm triggered by the infection, as discussed before.

Interestingly, while neurological manifestations are more common in patients with severe COVID-19 pneumonia [92], encephalitis has also been reported in patients with mild respiratory symptoms [83,93]. Considering this in mind, it is crucial to consider this complication in medical practice and conduct additional studies to uncover its mechanisms.

#### 3.1.2. Acute Hemorrhagic Necrotizing Encephalopathy

Acute hemorrhagic necrotizing encephalopathy (AHNE) is a rare condition, presumably caused by an immune-mediated response to a viral infection. It has predominantly been described in the pediatric population but can also occur in adults. The most common finding in neuroimaging is multifocal symmetrical lesions with hemorrhagic necrotic characteristics, primarily located in the thalamus and, to a lesser extent, in the white matter, brainstem, and cerebellum [94]. The first case of AHNE associated with COVID-19 was reported in March 2020 by Poyiadji et al. [94], and the total number of reported cases has increased since then [95,96,97,98,99,100]. Clinically, AHNE associated with SARS-CoV-2 is characterized by altered mental status, irritability, seizures, and neurological deficits. In all cases, it was accompanied by a respiratory condition, and in most cases, there were no prior comorbidities, except for a 51-year-old woman with a history of aplastic anaemia [97].

On the other hand, the onset of AHNE has also been reported in a 29-year-old patient following vaccination with an inactivated virus. Notably, there were no signs of respiratory compromise, a mutation in RANBP2 (previously associated with AHNE), or the massive production of cytokines through antibody-dependent enhancement (ADE). For this reason, the authors proposed RANBP2 mutation and ADE as contributing factors in the pathogenesis of AHNE in COVID-19 [101]. However, further research is needed to determine the relationship between vaccines and AHNE.

It is presumed that the cytokine storm characteristic of COVID-19 and the dysfunction of the blood–brain barrier would explain the progression of this complication. Thus, therapeutic approaches have been based on steroids, plasmapheresis, and immunoglobulins. However, the long-term benefit of these therapies remains unknown [27]. Due to this and the high potential for disability with devastating neurological outcomes, a more detailed investigation of this condition is required to identify the factors involved and achieve proper prevention and treatment.

#### 3.1.3. Stroke

The occurrence of strokes is considered a rare but alarming complication of coronavirus infection. Upon analyzing a series of two-hundred and fourteen infected patients, it was found that out of these, six had been admitted for a stroke, and two presented with hemiplegia without prior respiratory symptoms. In total, 2.8% of the patients experienced a stroke, but this percentage rose to 5.7% when considering only patients with severe respiratory infection (SRI). Regarding laboratory tests, patients with SRI exhibited more pronounced lymphopenia and higher levels of D-dimer compared to their counterparts without severe infection (18). The elevation of D-dimer has been associated with coagulation abnormalities and is linked to a more severe course of the disease, often observed in deceased patients [22,102,103].

Another case report, conducted in London, United Kingdom, included six patients diagnosed with COVID-19 and acute ischemic stroke. It was found that all of them presented markedly elevated levels of D-dimer. In two of them, the stroke occurred despite being under anticoagulant therapy. Additionally, five out of the six patients showed positive lupus anticoagulant, and one had medium-title anticardiolipin IgM and low-title IgG and IgM antibodies against glycoprotein-1 [84,104].

Despite the reported cases, the mechanisms behind COVID-19-associated cerebrovascular disease have not yet been fully elucidated. However, multiple mechanisms have been suggested, including the presence of a prothrombotic state related to the exacerbated immune response, which leads to the release of a large number of pro-inflammatory cytokines and chemokines (IL-6, IFNγ, MCP1, and IP-10) [105,106,107], resulting in an imbalance in the homeostasis of procoagulant and anticoagulant mechanisms characteristic of viral infections [108]. The interaction between pro-inflammatory mediators, the endothelial wall, and platelets, with subsequent endothelial dysfunction, activation, and Toll-like receptor activation, could generate this prothrombotic state [109]. In line with this, a study analyzed a group of patients admitted to the intensive care unit due to SARS-CoV-2 infection and found elevated levels of fibrinogen, C-reactive protein, coagulation factors VIII and von Willebrand factor. D-dimer was substantially elevated, and antithrombin was found to be marginally decreased, supporting the hypothesis that SARS-CoV-2 induces a state of severe inflammation at the expense of hemostasis, leading to a hypercoagulable state [110,111].

Furthermore, atherosclerosis may play a fundamental role in the development of acute events. The atheromatous plaque that tends to form with ageing [112] may destabilize due to the interaction between inflammatory cells and the endothelium, leading to the rupture of the capsule covering the atheroma and the release of a thrombus [113].

#### 3.1.4. Seizures

A seizure is a paroxysmal alteration of neurological function characterized by excessive and hypersynchronous discharges of cortical neurons [114]. Epileptic seizures can occur in various scenarios, such as provoked seizures in epileptic patients, with an incidence of up to 30%, or in critically ill non-epileptic patients due to SARS-CoV-2. In this case, it is an acute symptomatic epileptic seizure with an incidence of 1.1% [115]. However, there are case reports of patients without severe respiratory disease presenting seizures associated with a positive PCR for SARS-CoV-2 [85,115,116].

Several hypotheses have emerged, and it is presumed that among the neuropathological mechanisms involved in the presentation of seizures, the induced inflammatory state due to the infection plays a significant role [117], leading to a substantial release of cytokines (interleukin (IL)-1β, IL-6, tumor necrosis factor (TNF), and chemokines (C-C chemokine ligands (CCL)-2, CCL-3, CCL-5) [118]. The activity of these substances causes neuronal hyperexcitability through the glutamate pathway, predisposing to the initiation of epileptogenic activity [119].

Another involved mechanism is hypoxia. Alveolar hypoventilation leads to hypoxemia and consequent cerebral tissue acidosis, which is also associated with a high risk of epileptic seizures [120,121]. Additionally, the presence of seizures can occur in the context of a viral encephalopathy, as mentioned earlier, and has also been reported as an adverse effect related to the use of antiviral drugs [122].

#### 3.1.5. Delirium and Other Neuropsychiatric Manifestations

A meta-analysis examined 72 independent studies focused on the neuropsychiatric presentations of coronaviruses. Regarding COVID-19, it was found that, among patients in the ICU, 26 [65%] out of 40 exhibited dementia, 40 [69%] out of 58 showed agitation, and 17 [21%] out of 82 experienced altered levels of consciousness [123].

Furthermore, there is a report on three cases of COVID-19-positive individuals who did not exhibit the typical viral infection symptoms but instead simultaneously displayed acute episodes of anxiety, paranoia, disordered thinking, psychomotor agitation, and, in two of the cases, auditory hallucinations. All individuals studied exhibited elevated C-reactive protein levels, leading to the hypothesis that the aetiology is related to the virus’s potential for neuroinflammation. The literature has extensively described the relationship between immune-triggering factors and the development of neuropsychiatric disorders [124].

Based on previous research involving similar viruses, it has been suggested that the origin of neuropsychiatric symptoms may be attributed to the virus’s ability to invade the central nervous system (CNS), leading to an exacerbated release of cytokines and chemokines that ultimately result in inflammation within the CNS. Consequently, this affects the blood–brain barrier (BBB), allowing immune cells to migrate into the CNS, potentially causing neurotransmitter imbalances [125,126,127].

#### 3.1.6. Cephalea

Historically, cephalea has been associated with viral infections, particularly during epidemics and pandemics. The prevalence of cephalea associated with COVID-19 ranges from approximately 6.5% to 71%. It is essential to address it when it occurs as an acute manifestation and in a post-viral context [127].

The pathophysiological mechanism behind cephalea during recovery is believed to be linked to the systemic inflammatory response associated with fever and cytokine storm. This point is supported by increased serum levels of IL-10 in patients with SARS-CoV-2 infection during the presence of cephalea. This type of cephalea tends to be moderate to severe and displays migraine-like characteristics. However, in patients with a history of migraines, its presentation may differ from the usual phenotype [128].

On the other hand, post-viral cephalea is attributed to the direct invasion of the virus into the central nervous system through retrograde axonal transport, primarily via the olfactory nerve. This type of headache manifests in various phenotypes, and treatment should be tailored to each specific presentation [129].

#### 3.1.7. Demyelinating Diseases of the Central Nervous System

It has been found that in COVID-19 infection, IL-6 and IL-17 levels, in particular, are elevated, leading to immune dysregulation, which is directly related to the pathogenesis of multiple sclerosis. However, the direct relationship of this association has yet to be entirely elucidated [130].

A systematic literature review searching for case reports of demyelinating diseases concluded that encephalitis and encephalomyelitis were the most common manifestations [131]. Another systematic review found that the mean duration between infection and demyelinating event was 11.5 days in the neuromyelitis optica spectrum disorder (NMO-SD), six days in myelin oligodendrocyte glycoprotein antibody-associated disease (MOGAD), and 13.5 days in multiple sclerosis (MS). There was a positive association between the onset of demyelinating disease and relapses in previously diagnosed patients; in most cases, patients received corticosteroid management with a good response [132]. In children, acute disseminated encephalomyelitis (ADEM) can occur, with this being the most commonly described association with SARS-CoV-2 infection, followed by transverse myelitis (TM), MOGAD, and NMO-SD, conditions that are also associated with other types of viral agents [133,134].

#### 3.1.8. Acute Myelitis

COVID-19 has been associated with a wide variety of spinal cord manifestations, such as acute transverse myelitis, acute necrotizing myelitis, SARS-CoV-2-related myelitis, ADEM, NMO-SD, MOG-AD, hypoxic myelopathy, spinal cord infarction, and spinal epidural abscess [135]. A growing number of case reports of acute transverse myelitis have been presented in association with patients with COVID-19. A systematic review evaluated twenty reported cases from 14 different countries, and it was determined that the average time for myelitis symptoms to manifest was 10.3 days after the onset of respiratory symptoms. In cases where cerebrospinal fluid PCR was performed, the results were negative. Inflammatory cells were found in cerebrospinal fluid in 77.8% of the cases evaluated. After several weeks of follow-up, 90% of individuals had partially or fully recovered [136].

On the other hand, a 32-year-old patient presented to the emergency room with a history of high fever and flu-like symptoms, accompanied by a sudden loss of muscle strength in the lower limbs and sphincter dysfunction. Laboratory results showed an average white blood cell count, elevated D-dimer, and C-reactive protein. The COVID-19 PCR test was positive. An MRI revealed diffuse hyperintensity predominantly involving the grey matter of the cervical, dorsal, and lumbar spinal cord. Moderate inflammation of the cervical spinal cord was also observed [137].

Another 60-year-old patient who was initially hospitalized for COVID-19 without neurological symptoms was discharged and three days later developed progressive weakness in the lower limbs and urinary retention. Clinical examination revealed hypoesthesia below T9 and moderate spastic paralysis with a bilaterally positive Babinski sign. A spinal cord MRI revealed T2 hyperintensity of the thoracic cord, suggestive of acute myelitis. The aetiology is assumed to be intrinsically related to an aberrant immune response. Other studies have suggested a direct infection of the central nervous system by the virus [138].

### 3.2. Peripheral Nervous System

#### 3.2.1. Olfactory and Gustatory Disorders

With the onset of COVID-19 and its subsequent evolution into a pandemic, a wide variety of symptoms caused by SARS-CoV-2 began to be described. Among these, anosmia and ageusia have been reported in various patients. These manifestations can occur from 2 to 14 days after exposure to the virus, with the particularity that, unlike most viruses, SARS-CoV-2 generates these symptoms without being directly associated with rhinorrhea or nasal obstruction [20,80].

Consequently, researchers like Lechien et al. studied 417 patients with PCR-confirmed COVID-19 using the short version of the Questionnaire of Olfactory Disorders-Negative Statements (sQOD-NS). The results showed that 85.6% of the patients developed olfactory dysfunction, of which 76.9% had anosmia and 20.4% had hyposmia. Additionally, phantosmia occurred in 12.6% of cases and parosmia in 32.4%. On the other hand, gustatory disorders were present in 88.8% of cases [20].

However, 72.6% of patients regained olfactory function after the resolution of other symptoms within the first eight days. 22.5% still had isolated gustatory dysfunction, and 23.6% had both olfactory and gustatory dysfunctions. At the same time, they determined that there was a significant association between both disorders and that women were significantly more affected. Furthermore, olfactory dysfunction appears before the other symptoms in 11.8% of cases [20].

However, Lee et al. surveyed 3191 patients with COVID-19. They determined that only 15.3% presented anosmia or ageusia. In comparison, Mao et al. analyzed 214 patients and identified anosmia in 5.1% of cases and ageusia in 5.6%, revealing very heterogeneous results, which could be attributed to methodological differences between the studies [139,140,141]. Meta-analyses such as the one by Tong et al., who analyzed ten studies on olfactory dysfunction and nine on gustatory dysfunction in SARS-CoV-2 positive patients, demonstrated a prevalence of 52.73% and 43.93% for olfactory and gustatory disorders, respectively [80].

Now, another very relevant and under-studied point is the pathogenesis of these disorders, for which only some hypotheses are available. Among them, the effect of SARS-CoV-2 on ACE2 stands out, as it is the receptor for the virus. It is found in the tongue, nasal mucosa, and olfactory bulb [142]. Therefore, based on this information and studies conducted with previous coronaviruses, it is proposed that through this enzyme, SARS-CoV-2 may damage the nasal epithelium and the olfactory bulb, resulting in the loss of mitral cells and the glomerular layer, thus affecting the olfactory pathway [143,144]. In turn, the sense of smell and taste are intimately correlated so anosmia could lead to ageusia [142].

However, cases of isolated ageusia have been described, indicating that it has its pathogenesis, which could be linked to lingual ACE2, as these have a modulatory role in taste perception. It has even been shown that inhibiting this enzyme alters taste through the inactivation of sodium channels in taste receptors [145,146,147]. However, these pathways are still hypothetical, and other authors link these disorders to the virus’s effects on other elements, such as sialic acid receptors, through the N-terminal domains of the S protein of SARS-CoV-2. These domains may contain sialic acid-containing glycans that allow their recognition and binding to the receptor [55].

#### 3.2.2. Guillain–Barre Syndrome

Guillain–Barré Syndrome (GBS) is an acute demyelinating polyradiculoneuropathy characterized by the sudden onset of flaccid neuromuscular paralysis, with ascending motor weakness and sensory abnormalities. It is usually triggered following a viral or bacterial infection and has been associated with multiple epidemics and pandemics, such as those caused by H1N1, Zika, and coronaviruses, including SARS-CoV and MERS-CoV. Hence, notable cases of GBS have been reported during the SARS-CoV-2 pandemic [66,148,149,150].

Zhao et al. described one of the earliest cases of GBS associated with COVID-19. The patient was a 61-year-old woman who had returned from Wuhan four days before seeking medical attention due to acute weakness in both legs. Neurological examination revealed symmetric weakness and areflexia in the legs and feet, which continued progressing. Within three days, it was accompanied by symmetric weakness in both arms and hands and diminished distal light touch and pinprick sensation. Subsequently, on the fifth day of her hospital course, she was diagnosed with GBS. On the eighth day, she developed a dry cough and a fever of 38.2 °C. A chest CT scan revealed ground-glass opacities in both lungs, and an RT-PCR assay on the oropharyngeal swab yielded positive results for SARS-CoV-2 [151,152].

Currently, more similar cases have been documented, with patients exhibiting neurological symptoms characteristic of GBS, which typically occur 5 to 10 days after the diagnosis of COVID-19 [149]. However, as noted by Juliao et al., the number of GBS and COVID-19 cases found in the literature is low. Therefore, they compiled data from 11 cases across seven countries and described that out of the eleven patients, eight were male and three were female. Among them, a 23-year-old male patient stood out, while the other cases ranged from 54 to 77 years [86]. GBS associated with COVID-19 can manifest in various clinical forms.

Several case reports have identified patients with classical Guillain–Barré, paraplegic variant, facial diplegia, Miller Fisher type, or cervicobrachial type, which can occur in isolation or as combined variants. The classical variant is the most frequently observed type of manifestation [86,153,154]. The patients showed elevated protein levels without an increase in cells in the cerebrospinal fluid, as is characteristic of this syndrome. It is worth noting that there are reports of cases where PCR testing for COVID-19 in cerebrospinal fluid yielded negative results [86,153,154].

The diagnosis was established based on clinical presentation, evidence of albumin-cytological dissociation in cerebrospinal fluid, nasopharyngeal COVID-19 testing, and electrodiagnostic tests. These tests primarily showed low compound muscle action potential amplitudes, delayed distal motor latencies, and absent F waves in the affected limbs. Additionally, electromyography revealed decreased recruitment [155,156]. These findings align with the proposed pathophysiology for developing GBS due to SARS-CoV-2 infection, indicating a potential mechanism of autoimmune cross-reactivity due to the virus’s ability to stimulate inflammatory cells and produce pro-inflammatory cytokines [18]. However, due to the absence of serological tests for antibodies against SARS-CoV-2, it is still unclear whether the virus induces the production of antibodies against specific gangliosides associated with this syndrome [157,158].

On the other hand, the therapeutic approach followed in most cases was based on established recommendations for GBS treatment, i.e., intravenous immunoglobulin for five days (400 mg/kg/day) or plasma exchange in patients who required it [153,159,160]. However, the response was not uniform across all cases. Some showed favorable progress and rapid restoration of mobility, while others progressed to flaccid paraplegia with minimal movements in some limbs, necessitating physiotherapy. Some patients required mechanical ventilation and admission to the intensive care unit [152,161]. In addition, despite no predisposing factors being identified for the development and severity of the syndrome, Arnaud et al. and Alberti et al. emphasized that these patients had comorbidities with type 2 diabetes mellitus and developed severe respiratory failure [155,162].

## 4. Psychiatric Manifestations

In multiple studies, the development of psychiatric disorders such as depression, anxiety, post-traumatic stress disorder, delirium, psychosis, and neurocognitive alterations, among others, has been documented in individuals affected by COVID-19, either during their illness or in their recovery stage [163].

As a result of the neuroinflammation associated with COVID-19, systemic manifestations are triggered, including those within the nervous system. These may contribute to neuropsychiatric symptoms. Inflammatory cytokines such as TNF-α and IL-6 are the primary proposed mediators of this inflammatory response. Coupled with oxidative stress, they affect the integrity of neuronal cells and may lead to neuronal dysfunction, subsequently disrupting the neurotransmitter signaling of serotonin, dopamine, norepinephrine, and glutamate. This mechanism is considered the pathophysiological basis for neuropsychiatric alterations in these patients, such as mood dysregulation and cognitive impairment [164].

Additionally, in patients with COVID-19, changes in brain structure have been observed in neuroimaging, including volume loss and atrophy in some brain regions. However, more studies are needed to identify therapeutic targets that allow us to prevent and treat psychiatric manifestations in this specific context [165].

### 4.1. Anxiety

Anxiety is an unpleasant sensation, often accompanied by neurovegetative manifestations such as excessive sweating, accelerated heart rate, chest pressure, alterations in concentration and attention, and sometimes restlessness and agitation [166]. The concept of anxiety, in a general sense, encompasses various conditions, including generalized anxiety disorder, post-traumatic stress disorder, separation anxiety, panic disorder, social anxiety disorder, and other similar conditions [167].

A neuroendocrine stress response may exacerbate the psychological sequelae of COVID-19. Uncertainty, fear of contagion, and health concerns likely trigger cortisol release, potentially leading to amygdala hyperactivity. Heightened amygdala activity, a key structure in emotional processing and threat perception, could contribute to the persistence of anxiety symptoms. It has been observed that COVID-19 infection leads to decreased serotonin levels, dysregulation of noradrenaline release, functional alteration of GABA, and, consequently, a sustained inflammatory response. Changes in this neurotransmitter influence mood regulation and contribute to anxiety symptoms. Additionally, genetic predisposition plays a role in the stress response and increases the likelihood of developing an anxiety disorder. Therefore, additional genetic and environmental factors are involved in addition to those associated with COVID-19 infection [165].

### 4.2. Depression

The COVID-19 pandemic has led to a 25% increase in the global prevalence of anxiety and depression. In depression, there are disruptions in the regulation of noradrenaline, dopamine, and serotonin, which constitute the monoaminergic transmission system. The clinical manifestations are caused by alterations in the functioning of these systems, so therapies aimed at mood disorders focus on modulating one or more of these neurotransmitters. In COVID-19 infection, alterations in these neurotransmitters have been observed, as well as a marked decrease in the sensitivity of B receptors, which recognize antigens, such as pathogen proteins. These receptors trigger an immune response by binding to antigens and activating the production of antibodies to combat infections [168].

The genetic theory is complex and does not follow a classic pattern of inheritance but rather a susceptibility threshold model. Alterations have been found in the protein that acts as a transcription factor CREB1 on chromosome 2, which interferes with transcription factors, and anomalies on chromosome 17 that encode the SERT gene, which has a significant association with depression and stress. The prevalence of depression with medical comorbidities, such as COVID-19, increases the inflammatory burden within the CNS, which is associated with biochemical and neuroendocrine alterations, including the activation of the hypothalamic–pituitary–adrenal (HPA) axis and pro-inflammatory mediators increase like interleukin-1 (IL-1) [169].

### 4.3. Post-Traumatic Stress Disorder

Post-traumatic stress disorder (PTSD) is defined by exposure to a traumatic event where the individual is present, witnesses harm to others or receive information about a traumatic incident. It manifests with symptoms such as re-experiencing the event, avoiding thoughts and emotions related to it, reduced emotional expression, and signs of neurovegetative activation. The decrease in serotonin levels, dysregulation of noradrenaline release, and functional alteration of GABA lead to a considerable increase in post-traumatic stress symptoms in COVID-19 survivors and frontline healthcare workers responsible for managing such patients [167].

### 4.4. Psychosis

Psychosis is described as a significant alteration in sensory perception and thinking, leading to a loss of contact with reality. It is important to note that it is not a disorder itself but a symptom that may be related to various medical conditions or mental disorders leading to hallucinations (perceiving things that are not present) and delusions (strongly held false beliefs). It should be distinguished from delirium, an acute mental confusion caused by underlying medical factors, such as infections or metabolic imbalances [170].

During the stay in the intensive care unit, especially during severe illnesses, there is a higher risk of experiencing psychotic symptoms. These symptoms may include alterations in sensory perception like hallucinations, delusions, thought disorganization, and attention disturbances. The emotional tension and anxiety associated with hospitalization in an intensive care unit can also contribute to the onset of psychotic symptoms [24].

### 4.5. Cognitive Impairments

It has been reported that cognitive impairments manifested weeks after acute COVID-19; it is referred to as “brain fog”. This feature describes a feeling of mental confusion, difficulty concentrating, memory problems, and difficulty articulating words, among other symptoms. This alteration is usually seen in patients with long-COVID-19 and the associated inflammatory burden [171].

In addition, a series of studies conducted on post-COVID patients have highlighted the increase of up to one million working-age adults reporting significant problems with memory, concentration, or decision-making following the pandemic [172]. Similarly, it is estimated that patients who suffered from COVID-19 and required ICU admission experienced a loss of up to nine points in their intelligence quotient (IQ), while those with unresolved persistent symptoms lost six points, and those with mild COVID-19 and resolved symptoms lost only three points. However, reinfections resulted in an additional two-point loss in IQ. Longer hospital stays and the duration of acute illness were identified as the main predictors of persistent global deficits [173].

Nevertheless, cognitive impairment has also been reported in non-hospitalized patients, with “brain fog” being one of the most frequent neurological symptoms, surpassing anosmia in some studies. It affects women more frequently than men. Additionally, both hospitalized and ambulatory patients have shown various hypometabolic areas in the olfactory convolution, amygdala, hippocampus, cingulate cortex, pons, and cingulate cortex when undergoing fluorodeoxyglucose (FDG) positron emission tomography (PET). This suggests that this brain imaging study could be a valuable tool for diagnosing brain damage caused by prolonged SARS-CoV-2 infection and subsequent cognitive impairment [174].

This impairment could be explained by the fact that stress and anxiety related to the COVID-19 pandemic similarly affect the cytokine system and cholinergic pathways, resulting in cognitive alterations, while also causing the hyperphosphorylation of tau protein and activation of PKR kinase, both of which are proteins that accumulate abnormally in the brain and have been associated with the pathophysiology of Alzheimer’s disease [174].

In addition to this, other mechanisms have been proposed through which cognitive impairment would occur, including direct cellular injury by the virus, fusion between neurons and glial cells, increased microglial reactivity and reduced neurogenesis, impairment of the hypothalamic–pituitary axis, with a consequent cortisol deficit, microthrombosis, due to endothelial injury and platelet and complement activation, and even intestinal dysbiosis with altered tryptophan absorption, resulting in dysfunction induced by low levels of serotonin in vagal signaling [172].

In neuropsychiatry, research is being conducted to identify biomarkers that can be useful in measuring an individual’s physical and mental resilience to stressful situations. These biomarkers could serve as early indicators of cognitive impairments and will facilitate timely interventions in preparation for similar health events in the future. In this way, we can anticipate the development of a new generation of therapies focused on biological mechanisms to prevent and reverse neurodegenerative processes [175].

## 5. Conclusions

Despite COVID-19 primarily affecting the respiratory system, neurological complications have gained significance due to the increasing number of reported cases, along with the severity of their manifestations and sequelae. Consequently, it is important to elucidate its pathophysiology, which is debated between direct viral invasion and aberrant activation of the immune system. These mechanisms can impact both the central and peripheral nervous systems, giving rise to a wide array of neurological manifestations ranging from anosmia and ageusia to encephalitis, strokes, delirium, and Guillain–Barré syndrome, among others. Additionally, psychiatric disorders such as anxiety, depression, and psychosis have been observed, representing a significant health risk for patients with COVID-19. Therefore, it is crucial to consider the potential presence of these complications in patients to achieve early diagnosis and treatment, ultimately improving their prognosis. Similarly, further research and a comprehensive understanding of these neurological implications are imperative for developing effective management strategies.

## Figures and Tables

**Figure 1 biomedicines-12-01147-f001:**
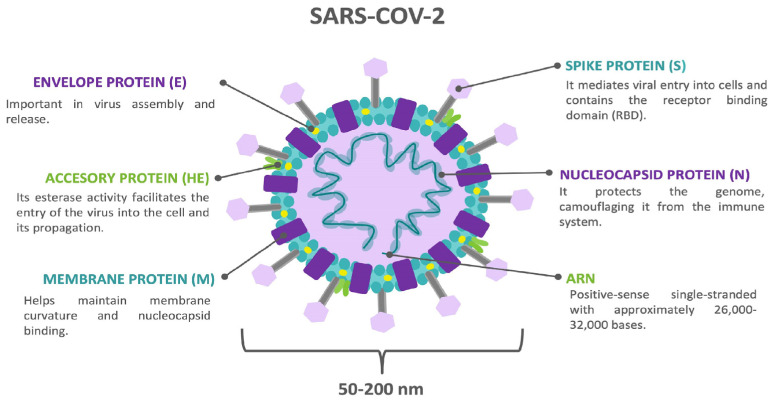
The SARS-CoV-2 structure.

**Figure 2 biomedicines-12-01147-f002:**
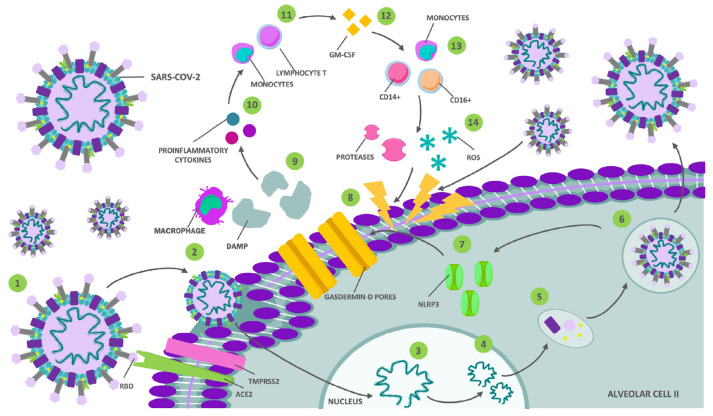
Mechanism of SARS-CoV-2 infection. (1) The interaction of the receptor-binding domain (RBD) of the S protein with the angiotensin-converting enzyme 2 (ACE2), as well as the action of proteases like trans-membrane serine protease 2 (TMPRSS2), which, together, allow (2) the entry of the virus into the type II alveolar cell. Subsequently, (3) the viral RNA enters the cell nucleus for (4) replication and (5) protein biosynthesis, (6) both products are included in new viral particles that will be released. This results in (7) the activation of inflammasome 3 (NLRP3), (8) undergoing cell apoptosis and pyroptosis through gasdermin-D pores, and (9) cytosolic content release, including some damage-associated molecular patterns (DAMPs) that are detected by alveolar macrophages, triggering (10) an inflammatory response mediated by pro-inflammatory cytokines that (11) attract monocytes and T lymphocytes. The latter (12) releases granulocyte–macrophage colony-stimulating factor (GM-CSF), which (13) activates CD14+ and CD16+, resulting in a pro-inflammatory feedback loop that ultimately (14) leads to severe lung damage through the direct action of the virus, proteases, and reactive oxygen species.

**Figure 3 biomedicines-12-01147-f003:**
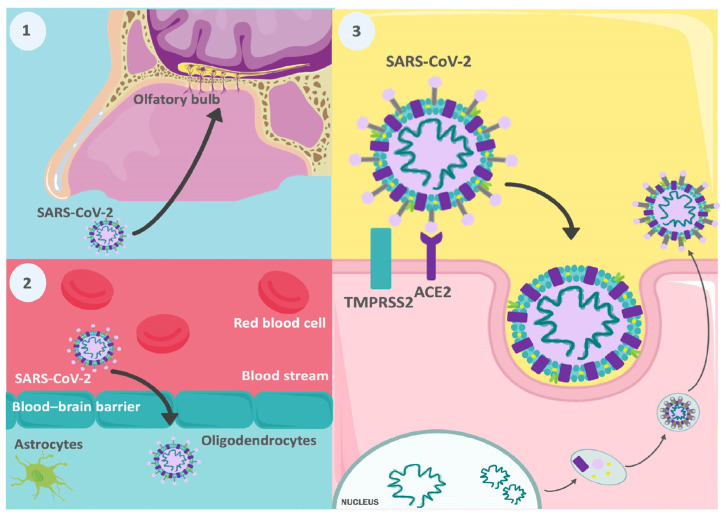
Potential neuroinvasive pathways of SARS-CoV-2. (**1**) It has been speculated that SARS-CoV-2 can invade the brain through olfactory pathways. (**2**) Hematogenous spread is another possible form of neuroinvasion; some similar viruses have been able to cross the blood–brain barrier. (**3**) The virus can invade tissues, utilizing TMPRSS2 and ACE2 receptors. Abbreviations: TMPRSS2, transmembrane protease, serine 2; ACE2, angiotensin converting enzyme 2.

**Figure 4 biomedicines-12-01147-f004:**
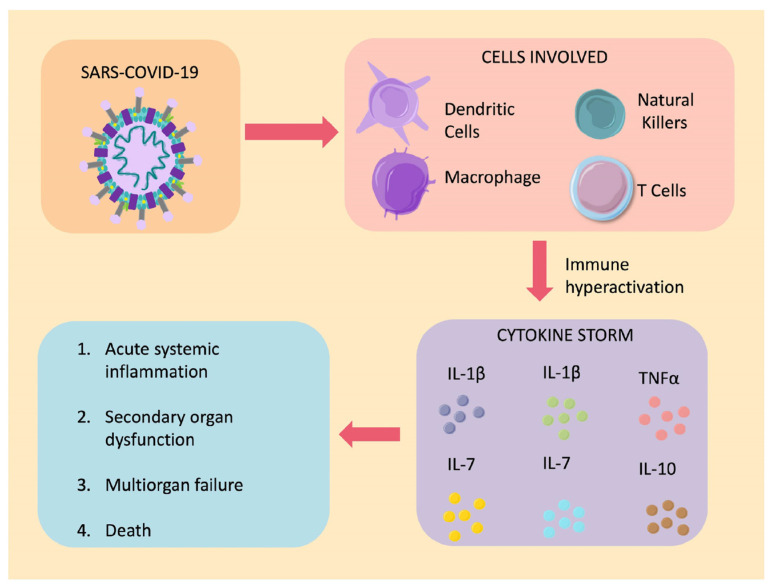
Cytokine storm in SARS-CoV-2 infection. Patients infected with SARS-CoV-2 present immune dysregulation, characterized by hyperinflammatory events. First of all, there is an increased activity of a wide variety of immune cells, such as dendritic cells, macrophages, T cells and natural killer cells, leading to a significant elevation in plasma concentrations of interleukin-1β (IL-1β), interleukin-7 (IL-7), interleukin-10 (IL-10), interferon-gamma (IFNγ), tumor necrosis factor α (TNFα), and vascular endothelial growth factor (VEGF), ending in dysregulation of the immune system leading to acute systematic inflammation, secondary organ dysfunction, multi-organ failure and finally death in many cases [61].

**Table 1 biomedicines-12-01147-t001:** Manifestations and neurological complications of COVID-19.

Manifestation/Complication	Authors	Study Type	Description of the Study/Case	Results
** *Olfactory and gustatory disorders* **	Tong et al. (2020) [80]	Systematic review and meta-analysis	The literature search included articles on the prevalence of olfactory or gustatory dysfunction in patients with COVID-19.	Ten studies were analyzed for olfactory dysfunction, while nine were examined to determine gustatory dysfunction. Subgroup analyses were performed for studies assessing olfactory dysfunction using validated and non-validated instruments. It was found that olfactory dysfunction had a prevalence of 52.73% and gustatory dysfunction of 43.93%. In general, studies reported a higher prevalence of olfactory dysfunction.
	Lechien et al. (2020) [20]	Multicenter epidemiological study	Clinical data from patients with laboratory-confirmed COVID-19 infection have been collected from four Belgian hospitals. The clinical data was prospectively gathered. A total of 417 patients completed the study.	Among the general symptoms, cough, myalgia, loss of appetite, diarrhoea, fever, headache, and asthenia were the most prevalent, representing more than 45% of the patients. A total of 85.6% of the patients had olfactory dysfunction related to the infection. Phantosmia and parosmia affected 12.6% and 32.4% of the patients, respectively. Likewise, 88.8% reported taste disorders.
** *Neuropsychiatric manifestations* **	Helms et al. (2020) [81]	Case series	An observational series of 58 out of 64 consecutive patients admitted to the hospital for COVID-19.	Neurological findings were recorded in 8 out of 58 patients (14%) upon admission to the ICU (prior to treatment) and in 39 patients (67%) when sedation and neuromuscular blockade were discontinued. Among the observed neuropsychiatric manifestations are delirium (26/40), agitation (40/58), signs of the corticospinal tract (39/58), and dysexecutive syndrome (14/39).
** *Myalgias* **	*Yang* et al. (2020) [82]	Retrospective observational study	Observational, retrospective, was performed at Wuhan Jin Yin-tan Hospital, where the clinical courses and outcomes of 52 critically ill patients were evaluated.	The most frequent symptoms were fever (98%), cough (77%) and dyspnea (63.5%). Myalgia was present in 11.5% of patients.
** *Encephalitis* **	Pilotto et al. (2020) [83]	Case Report	A 60-year-old man presented with severe alteration of consciousness, followed by fever, cough and cognitive fluctuations. CR of nasopharyngeal swab confirmed SARS-CoV-2 infection. He was treated with methylprednisolone 1 g/day for five days.	The cerebrospinal fluid PCR was negative for SARS-CoV-2, although inflammatory findings were present, with mild lymphocytic pleocytosis (18/μL) and a moderate protein increase. Additionally, the EEG showed generalized slowing, with decreased reactivity to auditory stimuli. After completing the 5-day therapy, the patient showed significant improvement. Upon discharge, on the 11th day of admission, the patient presented a normal neurological examination.
	Moriguchi et al. (2020) [44]	Case Report	A 24-year-old man with altered consciousness, convulsions, and neck stiffness, followed by a headache, generalized fatigue, fever, and a sore throat. PCR detected SARS-CoV-2 in CSF.	CMR indicated right lateral ventriculitis and encephalitis, mainly in the right mesial lobe and hippocampus. After admission to the ICU, empirical treatment with EV ceftriaxone, vancomycin, acyclovir, steroids and favipiravir was started. However, his evolution was not reported.
** *Stroke* **	Mao et al. (2020) [22]	Retrospective observational study	Multicenter, observational, retrospective study. In this study, 214 patients with COVID-19 were evaluated for severe (88) and non-severe (126) conditions.	A total of 78 patients (36.4%) had neurological complications. Patients with severe infection were more prone to these complications, were also older and had more underlying disorders. The prevalence of stroke was higher in patients with severe pneumonia (5.7%) compared to those with mild pneumonia (0.8%).
	Beyrouti et al. (2020) [84]	Case series	The cases of six patients from the National Hospital for Neurology and Neurosurgery, Queen Square, London, UK, with acute ischemic stroke and COVID-19 are described.	All six patients had large vessel occlusion with markedly elevated D-dimer levels (≥1000 μg/L). Likewise, stroke occurred in two patients despite therapeutic anticoagulation. The findings suggest that ischemic stroke related to COVID-19 infection may occur in the context of a highly prothrombotic systemic state.
** *Seizure* **	Karimi et al. (2020) [85]	Case Report	A 30-year-old woman previously healthy with tonic-clonic seizure whose respiratory specimen was positive for COVID-19 using PCR.	CMR was normal, and a chest CT revealed focal ground-glass opacities. The patient’s symptoms improved with anticonvulsants and antiviral medications.
** *Guillain–Barré Syndrome* **	Juliao Caamaño and Alonso Beato (2020) [86]	Case Report	A 61-year-old patient diagnosed and treated as SARS-CoV-2 infection with pneumonia (hydroxychloroquine and lopinavir/Ritonavir for 14 days). After symptoms disappeared, he presented bilateral facial nerve palsy.	No other neurological findings on examination were present. A chest x-ray showed significant improvement in pneumonia. He was diagnosed with variant GBS and treated with low-dose oral prednisone; at two weeks, he started with barely noticeable improvement on both sides.

**Abbreviations:** RT-PCR: Reverse transcriptase polymerase chain reaction; PCR: Polymerase chain reaction; CSF: cerebrospinal fluid; EEG: electroencephalogram; CMR: magnetic resonance imaging; ICU: intensive care unit; EV: endovenous; CT: computed tomography; GBS: Guillain–Barré syndrome.

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
