# Peer review of "COVID-19: Unveiling the Neuropsychiatric Maze—From Acute to Long-Term Manifestations"

_biomedicines, 2024, doi:10.3390/biomedicines12061147_

Round 1
Reviewer 1 Report
Comments and Suggestions for Authors
SARS-Cov2 infection is associated to a wide range of symptoms, including neurological complications, such as encephalitis, seizures, and Guillain-Barré syndrome. The aim of this review was to explore the various mechanisms through which the virus impacts the nervous system.
The review provides a comprehensive compilation of literature on the topic of neurological complications associated to COVID-19, enhanced by clear figure and table. However, some refinements are needed to improve the clarity and focus of the manuscript:
-Introduction: This section should be concise, more targeted, and specifically tailored to address the neurological impacts of COVID-19. Reducing general discussions and emphasizing key issues relevant to the scope of the paper will make the introduction more effective.
-Bradykinin Storm (Section 2.3): The inclusion of a separate paragraph on the Bradykinin Storm warrants reevaluation. If existing studies do not show a clear link between the Bradykinin Storm and COVID-19-associated neurological complications, it may be more appropriate to integrate this discussion into the broader context of the abberrant immune responses, rather than treating it as a distinct topic.
-Muscular Complications (Section 3.3): This section appears to diverge from the focus of the review. Since muscular complications are not necessarily neurological, reconsidering their inclusion or clarifying their relevance to the nervous system impact of COVID-19 could enhance the coherence of the review.
Comments on the Quality of English LanguageModerate editing required
Author Response
Thank you very much for taking the time to review this manuscript. Please find the detailed responses below and the revisions/corrections highlighted/in track changes in the re-submitted files.
A Point-by-point Response to Comments and Suggestions for Authors
Comment 1: -Introduction: This section should be concise, more targeted, and specifically tailored to address the neurological impacts of COVID-19. Reducing general discussions and emphasizing key issues relevant to the scope of the paper will make the introduction more effective.
Response 1: Thank you for pointing this out. We agree with this comment. Therefore, we have summarized this section by making it more concise, reducing discussion, and emphasizing key points. Page number 3, paragraphs 1-2, and lines 81-93.
Comment 2: -Bradykinin Storm (Section 2.3): The inclusion of a separate paragraph on the Bradykinin Storm warrants reevaluation. If existing studies do not show a clear link between the Bradykinin Storm and COVID-19-associated neurological complications, it may be more appropriate to integrate this discussion into the broader context of the aberrant immune responses, rather than treating it as a distinct topic.
Response 2: Thank you for pointing this out. We agree with this comment. The subsection 2.3. Bradykinin Storm was removed, and its related information was incorporated into section 2.2. Aberrant Activation of the Immune System. Additionally, the wording of that paragraph was slightly revised. Page number 8, paragraph 4, and lines 259-260.
Comment 3: Muscular Complications (Section 3.3): This section appears to diverge from the focus of the review. Since muscular complications are not necessarily neurological, reconsidering their inclusion or clarifying their relevance to the nervous system impact of COVID-19 could enhance the coherence of the review.
Response 3: Thank you for pointing this out. We agree with this comment. Therefore, we have eliminated the paragraphs related to muscle complications.
4. Response to Comments on the Quality of English Language
Point 1: Moderate editing of the English language required
Response 1: Thank you for pointing this out. We agree with this comment. Therefore, we have reviewed and improved the English Language.
Reviewer 2 Report
Comments and Suggestions for Authors
Authors review very well about the neurological manifestations after COVID-19 infection.
I think that this review has enough value to be published in Biomedicines to understand the long COVID-19.
Comments to the authors.
Major comments:
Fig. 1; I am not sure whether Fig.1 is necessary or not in this review. It is depending on Editor’s decision.
I think that the cognitive dysfunction in Long COVID may be more emphasized in section 4. Psychiatric Manifestations. “Cognitive impairment” is included in “4.5. Other Neuropsychiatric Disorders”. Please refer “Ziyad Al-Aly, Clifford J Rosen. N Engl J Med. 2024 Feb 29;390(9):858-860. doi: 10.1056/NEJMe2400189. Long Covid and Impaired Cognition - More Evidence and More Work to Do”.
Minor comments:
L42-44 and L47-49;
The following sentences are repeated “These sequelae, encompassing post-COVID-19 syndrome, long COVID-19, and post-COVID-19 conditions, are characterized by their onset during or after the documented infection and persistence beyond 12 weeks from diagnosis.”
Author Response
Thank you very much for taking the time to review this manuscript. Please find the detailed responses below and the revisions/corrections highlighted/in track changes in the re-submitted files.
A point-by-point response to Comments and Suggestions for Authors
Comment 1: Fig. 1; I am not sure whether Fig.1 is necessary or not in this review. It is depending on Editor’s decision.
Response 1: Thank you for pointing this out. We have discussed this among the authors and decided to keep this figure because we consider it necessary since it provides basic information to the reader that will allow them to better understand the pathophysiological mechanisms explained throughout the review. Page 2.
Comments 2: I think that the cognitive dysfunction in Long COVID may be more emphasized in section 4. Psychiatric Manifestations. “Cognitive impairment” is included in “4.5. Other Neuropsychiatric Disorders”. Please refer “Ziyad Al-Aly, Clifford J Rosen. N Engl J Med. 2024 Feb 29;390(9):858-860. doi: 10.1056/NEJMe2400189. Long Covid and Impaired Cognition - More Evidence and More Work to Do”.
Response 2: Thank you for pointing this out. We agree with this comment. We have read the recommended article and expanded the information about cognitive impairment. Page number 18, paragraphs 5-6, and lines 670-685. Page number 19, paragraphs 1-3, and lines 686-699.
Comments 3: L42-44 and L47-49; The following sentences are repeated “These sequelae, encompassing post-COVID-19 syndrome, long COVID-19, and post-COVID-19 conditions, are characterized by their onset during or after the documented infection and persistence beyond 12 weeks from diagnosis.”
Response 3: Thank you for pointing this out. We agree with this comment and the repeated sentences were removed.
Reviewer 3 Report
Comments and Suggestions for Authors
The review entitled : ″COVID-19: Unveiling the Neuropsychiatric Maze—From Acute 2 to Long-Term Manifestations″ is clear, comprehensive and of relevance to the review topic. All aspects of neurological complications reported in patients with COVID-19 were addressed by the authors. Figures are of high quality and cited references are appropriated. The conclusions are coherent and supported by the listed citations.
There is a minor point to consider before accepting the review for publication concerns the introduction in which sentences 40 to 44 are repeated (lines 44 to 49)
Author Response
Thank you very much for taking the time to review this manuscript. Please find the detailed responses below and the revisions/corrections highlighted/in track changes in the re-submitted files.
Point-by-point response to Comments and Suggestions for Authors
Comment 1: There is a minor point to consider before accepting the review for publication concerns the introduction in which sentences 40 to 44 are repeated (lines 44 to 49).
Response 1: Thank you for pointing this out. We agree with this comment. We have removed the repeated sentences.
Round 2
Reviewer 1 Report
Comments and Suggestions for Authors
Issues have been addressed
Comments on the Quality of English LanguageMinor editing required